# Are Ingested or Inhaled Microplastics Involved in Nonalcoholic Fatty Liver Disease?

**DOI:** 10.3390/ijerph192013495

**Published:** 2022-10-18

**Authors:** Teresa Auguet, Laia Bertran, Andrea Barrientos-Riosalido, Blanca Fabregat, Beatriz Villar, Carmen Aguilar, Fàtima Sabench

**Affiliations:** 1Grup de Recerca GEMMAIR (AGAUR)—Medicina Aplicada (URV), Departament de Medicina i Cirurgia, Universitat Rovira i Virgili (URV), Institut d’Investigació Sanitària Pere Virgili (IISPV), 43007 Tarragona, Spain; 2Servei Medicina Interna, Hospital Universitari de Tarragona Joan XXIII, Mallafré Guasch, 4, 43007 Tarragona, Spain; 3Servei de Cirurgia, Hospital Sant Joan de Reus, Departament de Medicina i Cirurgia, URV, IISPV, Avinguda Doctor Josep Laporte, 2, 43204 Reus, Spain

**Keywords:** microplastics, nanoplastics, obesogens, NAFLD, microbiota

## Abstract

Nonalcoholic fatty liver disease (NAFLD) has emerged as the predominant cause of chronic liver injury; however, the mechanisms underlying its progression have not been fully elucidated. Pathophysiological studies have stated that NAFLD is significantly influenced by dietary and environmental factors that could participate in the development of NAFLD through different mechanisms. Currently, “plastic pollution” is one of the most challenging environmental problems worldwide since several plastics have potential toxic or endocrine disputing properties. Specifically, the intake of microplastics (MPs) and nanoplastics (NPs) in water or diet and/or the inhalation from suspended particles is well established, and these particles have been found in human samples. Laboratory animals exposed to MPs develop inflammation, immunological responses, endocrine disruptions, and alterations in lipid and energy metabolism, among other disorders. MPs additives also demonstrated adverse reactions. There is evidence that MPs and their additives are potential “obesogens” and could participate in NAFLD pathogenesis by modifying gut microbiota composition or even worsen liver fibrosis. Although human exposure to MPs seems clear, their relationship with NAFLD requires further study, since its prevention could be a possible personalized therapeutic strategy. Adequate mitigation strategies worldwide, reducing environmental pollution and human exposure levels of MPs, could reduce the risk of NAFLD.

## 1. Introduction

In the last century, the global production of plastics has exponentially increased due to its widespread use in industry, agriculture, and daily life; thus, some plastics end up polluting the environment [1,2]. It is believed that large plastic polymers are inert and are not absorbed by the intestinal tract due to their size, so they are excreted unmetabolized. However, upon entering the environment and in biological systems, plastics breakdown into small particles by different transformation processes. This destruction produces massive amounts of smaller plastic particles, including microplastics (MPs) (size < 5 mm) and nanoplastics (NPs) (size < 1000 nm), [3] that are water, terrestrial, or airborne pollutants [4,5,6]. It is worrisome that several studies have found these smaller plastic particles in food [7,8,9,10,11,12,13,14] and in animal tissue samples [15,16]. In addition, MPs can serve as a vector for environmental pollutants or plastic additives and cause exposure to hazardous substances [17,18]. In humans, the ingestion and inhalation of MPs [6] and their presence in human stool [19] have been described. In this sense, there is a growing concern about the presence of MPs in food and the possible health consequences of their intake by humans [20,21]; however, data are still scarce.

In this review, we will comment on the evidence that has been reported regarding the possible involvement of ingested and/or inhaled MPs and NPs in the onset and progression of NAFLD, as graphically represented in Figure 1.

## 2. Nonalcoholic Fatty Liver Disease Pathogenesis

Nonalcoholic fatty liver disease (NAFLD) is considered one of the most prevalent chronic liver diseases worldwide due to the rapidly rising prevalence of obesity, diabetes, hyperlipidaemia, and cardiovascular disease. According to estimates, NAFLD affects up to one-third of adults in several industrialized and developing nations [24].

The hepatic pathologies of NAFLD range from simple hepatic steatosis (SS) to nonalcoholic steatohepatitis (NASH) and can even developing into hepatic fibrosis, cirrhosis, or liver cancer [25]. NAFLD is thought to be the metabolic syndrome’s manifestation in the liver. In this sense, recently, metabolic associated fatty liver disease (MAFLD) is a new term that has been proposed by experts as a more appropriate umbrella term since the heterogeneity of the pathogenesis of metabolic fatty liver diseases and the inaccuracies in their terminology require a revaluation of NAFLD nomenclature [26].

NAFLD is caused by a number of different pathogenic causes, such as genetic and epigenetic elements, insulin resistance, hormones released from adipose tissue, and imbalanced dietary patterns [27]. During the development of NAFLD, patients are susceptible to a number of complications, including hypertension, atherosclerosis, and other illnesses [28,29]. Nonetheless, the precise pathogenesis of NAFLD has not yet been thoroughly defined, and NAFLD-specific therapies have not yet received approval. Hence, an effective and safe treatment for this disease is urgently needed. According to recent studies, potential drugs in trials primarily concentrate on fibrosis, oxidative stress, inflammation, apoptosis, and metabolic imbalance.

For these reasons, an in-depth understanding of the pathogenic mechanisms involved in this disease is very important.

### 2.1. The Two-Hit Hypothesis

In accordance with the “two-hit” theory, abnormalities in the metabolism of glucose and lipids cause an excessive build-up of fatty acids in the liver. In hepatocytes, fatty acids can play a significant harmful role by causing oxidative stress and inflammatory processes that advance hepatocyte damage. In this regard, the first “hit” is associated with lipid metabolism disruption. Insulin resistance and the deregulation of key adipocytokines are characteristics of this “hit.” Hepatic steatosis, endoplasmic reticulum stress, oxidative stress, hepatocyte inflammation, and fibrosis are all strongly linked with the second “hit” [30].

### 2.2. The Multiple-Hit Hypothesis

The “multiple-hit hypothesis” is thought to adequately describe the pathogenesis of NAFLD. According to this theory, a number of concurrently active variables cause NAFLD pathogenesis, including nutritional factors, proinflammatory diets, insulin resistance, oxidative stress, adipose tissue inflammation, adipokines, obesity, type 2 diabetes, hormones, the gut microbiota, and genetic and epigenetic factors [31,32,33]. Insulin resistance and hepatic free fatty acids (FFAs) are really involved in the mechanisms of NAFLD [34]. Specifically, glucose and fat metabolism imbalance induces an overload of FFAs. These FFAs enter liver cells and are transformed into triglycerides. Triglyceride accumulation in hepatocytes results in SS, or nonalcoholic fatty liver (NAFL), which is characterized by a fat accumulation in the liver exceeding 5% of its weight. Moreover, excess FFAs enhance endoplasmic reticulum stress and mitochondrial pressure. Consequently, this enhances the production of reactive oxygen species (ROS) in the liver, triggering inflammation. NASH is a pathological condition that causes lobular inflammation, portal vein inflammation, and hepatocyte damage. Some hepatocytes experience apoptosis or necrosis as NASH progresses, and these cells produce more inflammatory substances. To start the process of liver fibrosis, these substances stimulate hepatic stellate cells, inducing the hepatic accumulation of the proteins of the extracellular matrix, which promotes fibrosis [35]. Cirrhosis and liver cancer may then occur as a result of the evolution of this fact, which could require liver transplantation for treatment [36].

#### Intestinal Dysbiosis

It is becoming increasingly evident that the gut microbiome plays a vital role in the onset and progression of NAFLD through the gut-liver axis. This axis refers to the bidirectional relationship between the gut, along with its microbiota, and the liver, arising as a result of interactions between signals generated by dietary, genetic, and environmental factors [37]. On the other hand, it is well known that liver diseases can alter the gut microbiota, in a mutual relationship in which the two systems are able to influence each other. 

The gut microbiota is considered an indispensable organ that interacts with host cells during metabolic processes [38]. A balanced gut microbiota community is essential to maintain the homeostasis of body metabolism. In this regard, Backhed et al. reported that intestinal microbiota can regulate the harvesting and storage of energy from the diet [39].

An increase in the aberrant microbiota leads to an imbalance in the microbial community, which is called intestinal dysbiosis. Gut microbiota dysbiosis has been reported to be associated with many metabolic disorders, including NAFLD [23], by dysregulating the gut-liver axis. This leads to increased intestinal permeability and the unrestrained transfer of microbial metabolites into the liver. The gut-liver axis represents the bilateral communication between the intestine and the liver, and it is connected by portal circulation, the bile tract, and systematic circulation [40]. It is known that the intestinal barrier of the colon is composed of several layers of defence. Under healthy and homeostatic conditions, the most external layer of defence is the mucus, which is composed of an outer microbiota-colonised layer, and an inner sterile layer. The epithelium below is a monolayer of cells sealed one to the other by tight junctions. A further layer of defence is provided by the gut vascular barrier, which controls the systemic dissemination of microbial metabolites and the microbiota through the portal circulation. Moreover, the intestinal mucosal barrier is further reinforced by the presence of a series of immune cells that contribute to the establishment of the barrier. However, under inflammation, the intestinal barrier can be disrupted at several places; when the gut vascular barrier is also damaged as demonstrated by increased detection of the fenestrated marker PV1, then the translocation of microbes or inflammatory microbial metabolites can occur to systemic sites, including the liver where they can induce local inflammation and promote liver disorders, such as NAFLD. Moreover, the intestinal mucosal barrier is further reinforced by the presence of a series of immune cells that contribute to the establishment of the barrier [37,41].

Current data indicate that a high-fat diet alters the microbiome, which in turn impairs the intestinal barrier and the gut vascular barrier. Intestinally derived bacteria and their components can easily reach the liver through the portal vein. Indeed, evidence shows that gut dysbiosis is involved in the development and progression of liver steatosis, inflammation, and fibrosis seen in the context of NAFLD, as well as in the process of hepatocarcinogenesis. Several different mechanisms have proposed to explain the role of the gut microbiota in the pathogenesis of NAFLD, including a dysbiosis-induced increased intestinal permeability (the “leaky gut”), an increased dietary energy harvest, the regulation of choline metabolism, the production of short chain fatty acids (SCFAs), and the bile acids metabolism [42].

Regarding intestinal derived bacteria, in the last decade, different studies have sought to correlate microbiome signatures with NAFLD phenotypes using culture independent techniques [43,44,45,46,47,48,49]. Although some contradictory evidence has arisen, both animal and human research have found compositional changes in the gut microbiota in relation to NAFLD-spectrum illnesses. Indeed, a decreased variety of the microbiota population living in the gut has been linked to hepatic steatosis. Although varying results have been reported at the class, order, family, and genus levels, NAFLD is typically described as having a rise in Firmicutes and a reduction in Bacteroidetes [50]. Furthermore, NASH patients showed a smaller percentage of Bacteroidetes than both NAFL and healthy subjects, according to Mouzaki et al. [43]. However, it is unlikely that single microbiome signatures will completely explain the different NAFLD phenotypes. These phenotypes probably result from varying impacts of the different microbiome signatures on the host according to its genetic predisposition or environmental factors [37].

However, changes in the functional capacity of the gut microbiome are probably more relevant than changes in microbiota composition. In this sense, modifications to bacterial taxonomy might not be as crucial to the pathogenesis of NAFLD as modifications to bacterial genes, as seen in metagenomics and metatranscriptomics. Metabolites frequently serve as mediators in the intricate interactions between the host and the intestinal microbiome. Both bacterial products and metabolites derived from the actions of the gut microbiome on exogenous (from diet and environmental exposure) and endogenous (from bile acids and amino acids) substrates can reach the liver through the portal vein and promote inflammation and metabolic abnormalities. Once in the liver, bacterial products can stimulate hepatic immune cells, activate inflammation pathways, and eventually participate in the pathogenesis of NAFLD and NAFLD-related liver cancer [51]. As mentioned, the role of gut microbiome dysbiosis in NASH progression can be partially attributed to increased susceptibility to intestinal permeability, which encourages the insertion of microbes and/or microbial products (endotoxins, lipopolysaccharide (LPS), and peptidoglycans). In this sense, the serum levels of some proinflammatory substances derived from gut microbiota, such as LPS, are higher in NASH patients than in NAFL patients [52]. In the liver, LPS can activate toll-like receptor 4 (TLR4) [53]. TLR4 promotes liver macrophage ROS generation and increases the expression of pro-interleukin-1β. This further contributes to a proinflammatory environment that triggers NASH processes [54].

In addition, the composition of the gut microbiome determines the production of the secondary bile acids, and influences FXR-mediated signalling in the intestine and the liver [55,56]. Therefore, the main players, such as diet, microbiota, and intestinal mucosa, are interrelated and connected to the host through the bile and blood flow portal. Moreover, bile acids produced in the liver regulate microbiota composition and intestinal barrier function, and microbiota metabolites regulate bile acid synthesis and glucose and lipid metabolism in the liver [57].

Thus, with all these premises, it is easy to hypothesize that the gut-liver axis can provide a therapeutic target for NAFLD, which is important, since, to date, there is no approved specific treatment for this disease. Improving the gut-liver axis can protect the liver from pathogenic components from the intestine. In this sense, a number of interventions targeting the gut-liver axis are in development and can be systematized according to whether they target the intestinal content and mucus, the intestinal microbiome, the intestinal mucosa, or targets outside the intestine. Several different approaches targeting the gut homeostasis such as antibiotics, prebiotics, probiotics, symbiotics (a combination of both a prebiotic and a probiotic), metformin, bile acid homeostasis targeting, adsorbents, TLR-4 signalling, fibroblast growth factor 19 (FGF-19) and FGF-21 signalling, bariatric surgery, and faecal microbiota transplantation are emerging as promising therapeutic options. Studies on humans appear to confirm the significant effects of microbiome-targeted therapies that have been shown in mice models. Large clinical trials are currently being conducted to examine the metabolic effects of microbiome-targeted treatments in NAFLD [58,59].

For all this evidence, gut microbiome dysbiosis seems to be a crucial factor in the development of not only hepatic steatosis but also NASH/fibrosis. Therefore, it is very important to know any circumstance that could induce the development of intestinal dysbiosis, in order to avoid or eliminate it.

## 3. Nonalcoholic Fatty Liver Disease Treatment

The complexity of the various pathways involved in the development of NAFLD complicates the treatment of this disease. Given the strong association with obesity and type 2 diabetes mellitus, lifestyle changes and weight loss are major targets in treatment of NAFLD and NASH. According to the guidelines of the European Association for the Study of Liver, pharmacological therapy should be implemented in patients with progressive NASH or in patients with early NASH but with factors such as increased transaminases, metabolic syndrome, diabetes mellitus, and age over 50 years. Pharmacological options for NAFLD include antidiabetic or anti-obesity drugs, treatments that modify the lipid profile, and vitamin supplements. Although the increasing knowledge on pathophysiological mechanisms involved has led to a significant search and development of different drugs acting at different levels, up to the present, no specific NAFLD treatments have been approved by regulatory agencies [60]. However, currently, there are many ongoing clinical trials with different targeted drugs. Probably, in the near future, the drug combinations will allow targeting both metabolic dysfunction and liver damage, as well as designing a personalized pharmacological therapy according to patient characteristics [61].

## 4. Microplastics: Routes of Exposure and Mechanisms of Toxicity

Plastics used in consumer products and discarded in the environment undergo slow degradation by oxidative processes and biodegradation, resulting in fragmentation into pieces of less than 5 mm, called secondary MPs. There are also primary MPs that are intentionally produced to be used in cosmetics or in different industries. Although plastics can be useful for human health when used in the manufacture of certain medical equipment, it is clear that MPs became ubiquitous environmental contaminants leading to inevitable human exposure, with hypothetical health implications [62].

Regarding the routes of human exposure to MPs, several studies have detected them in food, air, drinking water, wastewater, cosmetics, textiles, and dust [21,63,64]. Therefore, the main routes of entry of these materials into the body are the gastrointestinal and pulmonary routes. In this sense, these materials could pose a potential risk to the human population [65]. Ingestion is considered the major route of human exposure to MPs [66], and these particles have been reported in different food items [10,67,68]. Humans are estimated to ingest tens of thousands to millions of MP particles annually, or on the order of several milligrams daily. Because MPs are similar in size to the food of many aquatic organisms, MPs are often ingested by these organisms by mistake [69]. In humans, MPs may reach the gastrointestinal system through contaminated foods (aquatic organisms that ingest MPs, contaminated water or milk, and table salt, among others) or through mucociliary clearance after inhalation. As mentioned, it is generally believed that large plastic polymers are inert and are not absorbed by the intestinal system. Once ingested, >90% of MPs were reported to be excreted in faeces [70], especially large particles >150 μm; however, smaller particles may be absorbed systematically. It has been reported that MPs 0.1–10 μm in size can cross the blood-brain barrier and the placenta [71], particles < 150 μm can cross the gastrointestinal epithelium, and particles < 2.5 μm can enter the systemic circulation through endocytosis [71,72]. Upon entering in biological systems, plastics break down into small particles through biotic and abiotic weathering and transformation processes, creating massive amounts of smaller plastic particles in the environment, including plastic particles of <5 mm or MPs. Oral ingestion is followed by a number of steps that influence the particles and therefore their interactions, such as the contact with digestive fluids, the contact to intestinal cells, uptake and transport in the intestine and liver, and excretion [73,74]. In order to know if the MPs that have entered the digestive system have the capacity to cause detrimental effects on health, different works have been carried out. In this line, the toxic effects of MPs have been investigated in numerous aquatic species and inflammation, genotoxicity, and oxidative stress responses have been described [15]. However, data on effects in mammalian systems are limited [21]. In this sense, studies using in vivo models have described different effects as the accumulation of MPs in tissues such as the intestine, kidney, and liver [16]; the decreased colonic mucus secretion [75]; a minor uptake in intestinal cells [76]; an increase of bile acids and their metabolites in the liver [77]; the intestinal barrier dysfunction [78,79]; some fatty acid and metabolic disorders [75,78,79]; and also the appearance of intestinal dysbiosis [78,79]. In the same sense, it has been suggested that these changes in gut microbiota composition could increase gut permeability, alter metabolism, and favour an inflammatory response [80]. It is worth mentioning here, that MPs may have a different behaviour if there are intestinal diseases that induce inflammation of the intestinal mucosa, although more studies are needed in this regard [74].

The second route of exposure is the inhalation of suspended MPs in the air. MPs are released into the air by numerous sources, including synthetic textiles, the abrasion of materials, and the resuspension of MPs deposited on surfaces [81]. Air MPs deposit in the respiratory system, and clearance by macrophages or migration to the circulation or lymphatic system may lead to particle translocation. The development of airway and interstitial lung disease in occupational exposure to airborne MPs has been described in workers of the synthetic textile, flock, and vinyl chloride or polyvinyl chloride industries [82,83,84]. This probably occurs under conditions of high concentrations or high individual susceptibility.

Finally, dermal contact with MPs is considered a minimal significant route of exposure, and it has been speculated that NPs (<100 nm) could also transverse the dermal barrier [85].

There is scientific evidence in several species of rodents and mammals, including humans, that some of the MPs reach the bloodstream and are distributed to different organs and tissues, in addition to inducing variations in blood parameters [86,87,88]. The presence of MPs/NPs in the lymph, the portal vein, cerebrospinal fluid, milk (in lactating animals and women), and the human placenta has been detected [21]. MPs and NPs were detected in the human placenta at both foetal and maternal faces and in chorioamniotic membranes [88]. In addition, MPs of approximately 10 μm were also found in lung tissues from autopsies [89]. The removal of MPs from systemic fluids has also been reported to occur through splenic filtration and urine excretion, and MPs have also been detected in human faeces [19,21].

With regard to the main mechanisms of toxicity, it has been suggested that they are oxidative stress and cytotoxicity [16,90,91], deregulation of energy homeostasis and metabolism [92,93,94], translocation to the circulatory system [95,96,97], immune dysfunction [98], neurodegenerative disorders [16], or serving as vectors of microorganisms [99]. Regarding this last mechanism, it is important to note that alterations to the gut microbiome due to MPs could lead to adverse effects, such as the proliferation of harmful species, an increase in intestinal permeability, and endotoxaemia [22]. All these MPs toxic mechanisms are intrinsically interconnected, so that disruption of one process can initiate a cascade of chained toxicological responses [100].

## 5. Microplastics as “Obesogens”

The two most commonly given explanations for obesity, genetics and energy balance, cannot fully explain the substantial increases in the incidences of obesity that have been observed worldwide [101,102,103]. Multiple environmental factors can affect obesity susceptibility such as gut microbiome composition, stress, and disrupted circadian rhythms, among others [104]. In this line, in 2003, the idea of linking endocrine-disrupting chemicals and obesity was introduced [105]. Humans are exposed to an ever-increasing number of environmental toxicants, some of which are important risk factors for metabolic diseases, such as diabetes and obesity. These metabolism-sensitive diseases typically occur when key metabolic and signalling pathways are disrupted, which can be influenced by the exposure to contaminants such as endocrine-disrupting chemicals. Metabolomics has allowed and continues to allow the identification of potential metabolic targets of endocrine-disrupting chemicals [106].

In this sense, it was demonstrated that certain endocrine-disrupting chemicals could activate nuclear hormone receptors that are important to the maturation of white adipocytes, such as peroxisome proliferator-activated receptor γ (PPARγ) [107], or interfere with hormonal signalling in the hypothalamus, pancreas, and liver [108,109]. In addition, they can produce epigenetic modifications, activate retinoid X receptor (RXR) or PPARγ, produce an increase in dysfunctional adipocytes, interfere with other signalling pathways, such as glucagon-like peptide 1 (GLP-1) or nuclear receptors that could alter metabolism, or even modify gut microbiota composition [104]. These endocrine-disrupting chemicals coin the term “obesogens” [110]. It is important to note that some of these “obesogens” contaminate different MPs. We would also like to emphasize that the effects of early-life “obesogen” exposure can be transmitted to future generations and that many epigenetic modifications have been observed after “obesogen” exposure [104].

## 6. Microplastics and Nonalcoholic Fatty Liver Disease

However, are the MPs truly related to NAFLD? If they are, through what mechanism could they act? Many studies have focused on the effects of MPs on marine organisms, and some reports have shown that MPs can enter the terrestrial food chain [111]. Moreover, it has been demonstrated that MPs interact with microbes in different media [112].

On the other hand, some studies have shown that some environmental chemicals, including antibiotics, pesticides, and several heavy metals, can effectively induce gut microbiota dysbiosis, change the mucus layer, and even result in lipid metabolism disorders in different experimental models, including mice [113,114,115,116,117]. In this sense, it has been reported that the intestine of mice fed high-concentration MPs showed obvious inflammation and higher TLR4 and interferon regulatory factor 5 (IRF5) expression. Thus, authors concluded that MPs can induce intestinal dysbacteriosis and inflammation [118]. Following the same line of research, Luo et al. investigated the impact and mechanism of MPs on dextran sodium sulfate (DSS)-induced colitis. The results demonstrated that gavage with MPs alone caused minimal effects on the intestinal barrier and liver status of mice. For mice with colitis, additional MPs exposure aggravated histopathological damage and inflammation, reduced mucus secretion, and increased the colon permeability. More importantly, MPs exposure also increased the risk of secondary liver injury associated with inflammatory cell infiltration [119].

More specifically, in relation to NAFLD (Table 1), in 2017 Deng et al. used fluorescent and pristine polystyrene microplastics (PS-MPs) particles with two diameters (5 μm and 20 μm) to investigate the tissue distribution, accumulation, and tissue-specific health risk of MPs in mice. They showed that MPs accumulated in the liver, kidney, and gut, have a tissue-accumulation kinetics and distribution pattern that was strongly dependent on the MPs particle size. In addition, the authors suggested that MPs exposure induced a disturbance of energy and lipid metabolism as well as oxidative stress [16]. Similar results have been reported in fish previously [90]. Subsequently, Lu et al. exposed male mice to two different sizes of polystyrene MPs. Their data showed that the polystyrene MPs could induce not only gut microbiota dysbiosis and hepatic lipid metabolism disorders but also decreased mucus secretion in the colon of these mice, providing new insights into the potential health risks caused by MPs in mammals [75]. Similarly, Li et al. aimed to investigate whether exposure to polystyrene-NPs has an impact on the development of NAFLD in chow- and HFD-fed mice. They demonstrated that polystyrene-NPs triggered the oxidative damage and proinflammatory response of hepatic tissue and accelerated the development of liver fibrosis in HFD-fed mice [120]. In this sense, increasing evidence indicates that the exposure of high-fat diet (HFD)-fed animals to other different pollutants, such as PM2.5, SiO2 NPs, or multiwalled carbon nanotubes, could also disrupt glucose tolerance and lipid metabolism, induce oxidative damage to hepatic tissue, and result in the occurrence of liver fibrosis [121,122,123]. It is also interesting to note that MPs serve as minor but efficient vectors for carrying other contaminants such as pesticides, pharmaceuticals, metals, and atmospheric pollutants (PM1, PM2.5), among others [124]. Therefore, it could be hypothesized that such interactions may alter the migration behaviour and exposure patterns of MPs, which could increase the impact on liver involvement.

It is important to point out here the results of the study of Huang et al. Although these authors do not investigate the relationship of MPs with NAFLD, they do investigate them in relation to insulin resistance and changes in the microbiota, which are two key pathogenic mechanisms in NAFLD. They evaluated the effects of polystyrene (PS) on insulin sensitivity in mice fed with normal chow diet (NCD) or high-fat diet (HFD) and explained the underlying mechanisms. Mice fed with NCD or HFD both showed insulin resistance after PS exposure accompanied by increased plasma LPS and pro-inflammatory cytokines such as tumor necrosis factor-α and interleukin-1β. Exposure to PS also resulted in a significant decrease in the richness and diversity of gut microbiota, particularly an increase in the relative abundance of Prevotellaceae and Enterobacteriaceae. Additionally, the experiment showed accumulation of MPs in the liver and other tissues and inhibition of the insulin signalling pathway in the liver of PS exposed mice. The authors suggest that the mechanism of PS exposure to induce insulin resistance in mice might be mediated through regulating gut microbiota and PS accumulation in tissues, stimulating inflammation and inhibiting the insulin signalling pathway [125].

On the other hand, as mentioned, recent studies have described the presence of MPs and NPs in human blood [71,72]. In this sense, the metabolic alterations, especially lipid ones, described in relation to the presence of MPs [16,75,124,126,127] could be indirectly related to the development of NAFLD. In other words, this circumstance should be considered among the possible mechanisms involved in the relationship between PMs and the development of NAFLD.

Nevertheless, to date, we have not found human studies examining the relationships among MPs, the microbiota, and NAFLD pathogenesis. It is clear that humans are potentially exposed to these pollutants and that the deregulation of the intestinal microbiota by MPs is related to NAFLD and metabolic diseases associated with insulin resistance in mammals. However, to assess this risk, it is necessary to monitor their exposure. In this sense, future studies should assess this risk in humans by monitoring exposure and their presence in, for example, faeces samples [6,19,21] to determine if changes in the microbiota are detected and if they are related to the severity of NAFLD. If verified, efforts to eliminate these pollutants would be urgently needed and should be included in the personalized treatment of NAFLD patients.

## 7. Conclusions

Growing consumption of plastic is leading to the increasing exposure of humans to MPs. There are several lines of evidence that suggest the potential involvement of MPs and NPs, which humans ingest and/or inhale, in the pathogenesis of metabolic diseases such as NAFLD, probably by modulating intestinal microbiota. This fact, that can be inferred from the literature, should be more deeply investigated since the prevention of the intake and/or inhalation of MPs and NPs or the modulation of the gut-liver axis or the intestinal microbiota could be a possible therapeutic strategy for NAFLD personalized treatment. All this evidence should lead to urgent adequate mitigation strategies worldwide, reducing environmental pollution and human exposure levels of MPs to reduce the risk of NAFLD.

## Figures and Tables

**Figure 1 ijerph-19-13495-f001:**
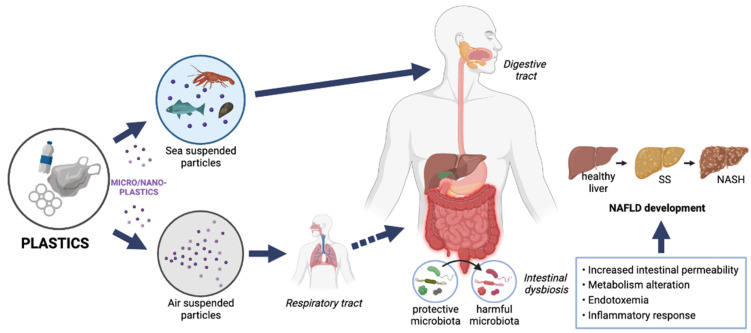
Microplastics and nanoplastics from plastic degradation enter organisms mainly through the digestive and respiratory systems (from this last system, microplastics pass to the digestive system by swallowing), producing changes in the gut microbiota that could induce intestinal dysbiosis. Intestinal dysbiosis triggers injuries (alterations of metabolic homeostasis, disruption of the intestinal barrier, endotoxaemia, and inflammatory response) that can cause health disorders, such as NAFLD [22,23]. NAFLD, nonalcoholic fatty liver disease; SS, simple steatosis; NASH, nonalcoholic steatohepatitis.

**Table 1 ijerph-19-13495-t001:** Summary of the reviewed literature using in vivo models to study the effect of micro- and nanoplastics on NAFLD.

Reference	Animal Model	MPs Type	MPs Size	MPs Administration Route	MPs Action
Deng et al., 2017 [16]	Male mice	Polystyrene MPs	Particles of two diameters (5 and 20 μm)	Oral	- Accumulation of polystyrene MP in the liver - Alteration of hepatic metabolism - Induction of energy and lipid metabolism disturbance and oxidative stress
Lu et al., 2018 [75]	Mice	Polystyrene MPs	Particles of two diameters (0.5 and 50 μm) at different concentrations	Oral	- Induction of gut microbiota dysbiosis and hepatic lipid metabolism disorders - Induction of decreased mucus secretion in the colon
Li et al., 2022 [120]	Experimental high fat diet (HFD)-induced mice	Polystyrene MPs	Different concentrations	Intravenous	- To trigger oxidative damage and proinflammatory response of hepatic tissue - To accelerate the development of liver fibrosis in HFD-fed mice

MP, microplastics; μm, micro meters; HFD, high fat diet.

## Data Availability

Not applicable.

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
