# Peer review of "Are Ingested or Inhaled Microplastics Involved in Nonalcoholic Fatty Liver Disease?"

_ijerph, 2022, doi:10.3390/ijerph192013495_

Round 1

Reviewer 1 Report

The authors presented an overview of recent research in which they try to investigate the role of microplastics and nanoplastics on nonalcoholic fatty liver disease (NAFLD).

The manuscript is well presented, so I have no major remarks regarding this manuscript. But, as this is a review paper, I would suggest the authors compile a table that gives a summary of current research and the references they have included in the paper. I find that It would help the readers in finding their way and give a better overview of the relevant research and results.

Author Response

First of all, thank you very much for being part of this review process. We appreciate so much your comments and suggestions that will help us to improve the quality of our manuscript. As you suggested, we have included a table (Table 1) with the main articles published in relation to the topic, that has allowed us to improve the corresponding section (lines 353-383, pages 8-9).

Reviewer 2 Report

This review collects evidence that has been reported regarding the possible involvement of ingested and/or inhaled microplastics/nanoplastics in the onset and progression of NAFLD. Overall, the findings are informative and provide some new knowledge, and therefore the article deserves to be published.

During the revision process of the manuscript, authors still need to be aware of the following issues.

  1. The authors need to add more discussion about the processes by which microplastics/nanoplastics enter the body by way of food chain transfer. There is still limited research on the bioaccumulation of microplastics/nanoplastics and in particular on the extent of biomagnification. Therefore, the reader is interested in the results of the entry into the human body by way of the food chain.
  2. A recent study found the presence of microplastics/nanoplastics in blood as well. We guide that if blood lipid levels increase in the blood, this may also indirectly affect the development of fatty liver disease. Therefore, the authors need to add more discussion on this interesting phenomenon.
  3. Studies have confirmed that long-term exposure to ambient air pollution may increase the chances of metabolic-related fatty liver disease. Have the authors considered the interaction between microplastics/nanoplastics and atmospheric pollutants (PM1, PM2.5, PAHs, etc.)? Such interactions may alter the migration behaviour and exposure patterns of microplastics/nanoplastics. More discussion by the authors is needed.
  4. More advice from the authors is needed on reducing environmental pollution and human exposure levels of micro/nanoplastics to reduce the risk of fatty liver disease.

Author Response

First of all, thank you very much for your contribution in this review process. Your comments and suggestions are really helpful.

1) In this sense, we have added more information about the processes by which microplastics/nanoplastics enter the body by food chain transfer and their possible toxic effects (lines 358-375, page 6).

2) In agreement with your suggestion, we have included a brief paragraph commenting on this possibility (lines 399-404, page 9).

3) We totally agree with you. In this sense, we have written a small explanatory paragraph in this regard (lines 376-380, page 8).

4) Accordingly, although it had already been mentioned in the body of the text, we have included a sentence to this effect in the Abstract (lines 30-32, page 1) and in the Conclusions sections (lines 422-424, page 9).

Reviewer 3 Report

The work has a good structure, but in my view it does not meet the minimum standards to be considered a good review on this topic. The article needs more data and tables. It is very brief.

Author Response

First of all, thank you very much to evaluate our manuscript. Your comments are really valuable and will help us to improve the quality of the article. In any case, we want to mention here that the evidences on this subject is scarce and, for this reason, the review is concise. However, we have made some changes in order to meet the standards.

In the first place, we have included a Table (Table 1) with the main articles published in relation to the topic, that has allowed us to improve the corresponding section (lines 353-383, pages 8-9).

Secondly, we have added more information about the processes by which microplastics/nanoplastics enter the body by food chain transfer and their toxic effects (lines 358-375, page 6).

Third, we have included a paragraph with information about the presence of MPs in the blood and its possible relationship with lipid metabolism and indirectly with the development of fatty liver disease (lines 399-404, page 9).

Finally, we have added information regarding the interaction between microplastics/nanoplastics and other atmospheric pollutants, which could facilitate the joint action of these molecules (lines 376-380, page 8).

Reviewer 4 Report

On account of the manuscript entitled ''Are ingested or inhaled microplastics involved in nonalcoholic 2 fatty liver disease? the authors evaluated''. The authors address in a review, evidence on the possible involvement of ingested and/or inhaled micro(nano)plastics in the onset and progression of Nonalcoholic fatty liver disease. The topic is important to conduct environmental risk assessment for the environmental contaminants including plastics since several plastics have potential toxic or endocrine disputing properties, and the authors got interesting results. The manuscript was well written and designed. After careful consideration, I made a decision that the manuscript is acceptable for publication in its present form.

Author Response

Answer: Thank you very much to review this manuscript. We appreciate so much your evaluation and your decision.